# Structural, Mineral and Elemental Composition Features of Iron-Rich Saponite Clay from Tashkiv Deposit (Ukraine)

**Halyna Sokol** [1,*], **Myroslav Sprynskyy** [2,*], **Alla Ganzyuk** [3], **Victoria Raks** [4] and **Bogusław Buszewski** [2]

1   RIOT Technologies Ltd., 50 B Czarnowiejska Str., Cracow 30-054, Poland
2   Department of Environmental Chemistry and Bioanalytics, Faculty of Chemistry,
    Nicolaus Copernicus University, 7 Gagarina Str., Torun 87-100, Poland; bbusz@chem.umk.pl
3   Department of Chemistry, Faculty of Technology Engineering, Khmelnitsky National University,
    11 Instytutska Str., Khmelnitsky 29016, Ukraine; alla.ganzyuk@gmail.com
4   Department of Analytical Chemistry, Faculty of Chemistry, Taras Shevchenko National University of Kyiv,
    60 Volodymyrska Str., Kyiv 01601, Ukrain; victoriia2005@gmail.com
*   Correspondence: sokolytko@gmail.com (H.S.); mspryn@chem.umk.pl (M.S.); Tel.: +48-786-224-818 (H.S.)

**Abstract:** Low-temperature nitrogen adsorption–desorption isotherms, scanning electron microscopy, transmission electron microscopy, X-ray diffraction, as well as infrared spectroscopy were used to characterize structural features of raw and acid-treated saponite from Tashkiv deposit of Ukraine. It was determined that raw saponite is predominantly composed of trioctahedral saponite with an admixture of dioctahedral nontronite and associated minerals such as quartz, hematite, and anatase. Raw saponite clay was characterized by a high content of iron (19.3%) and titanium (1.1%). Iron is present in the form of hematite particles, isomorphic replacements in octahedral and tetrahedral sheets of a clay structure, or as a charge-balancing cation in the interlayer space. Titanium is homogeneously dispersed as submicrometer anatase particles. The porous structure of both saponite forms consists of micro-meso porous system with narrow slit mesopores dominating. As a consequence of the acid treatment, the specific surface area increased from 47 to 189 $m^2$ $g^{-11}$, the total pore volume from 0.134 to 0.201 $cm^3$ $g^{-1}$, and the volume of the micropores increased sevenfold. Using the data of our research allowed us to utilize these mineral resources wisely and to process saponite more efficiently.

**Keywords:** saponite; acid treatment; morphology; nanocrystallite structure; porous structure

## 1. Introduction

Saponite is a trioctahedral phyllosilicate clay mineral that belongs to the smectite group, with the molecular formula, $(M_x^+ \times nH_2O)Mg_3^{2+}\left(Si_{4-x}^{4+}Al_x^{3+}\right)O_{10}(OH)_2$ where M represents the interlayer exchange cations [1]. It is a product of magnesium silicates that is commonly formed during the weathering process of basic volcanic materials [2].

One of the biggest industrial deposits of saponite clay mineral in Europe is found in the north-western part of Khmelnitsky region in the Slavuta district of Ukraine. These reserves contain over 100 million tons of saponite [3]. Unfortunately, their main characteristics important for industrial applications have been insufficiently studied, and the works previously done in this field are not complex [3–5]. Saponite deposits located in countries such as Canada, the United States, Japan, Turkey, the United Kingdom, Spain, and the Czech Republic [6–15] are well-known. However, the industrial applications of saponite clays from different locations are varied, because of distinctions in their

physical and chemical properties, which are dependent on structure and composition. Mineralogical composition of the raw clay samples which have been identified as saponites distinctively depends on the location of their deposit [5,13,16,17]. Mineral impurities within saponite raw samples usually consist of quartz, feldspar, clay (illite, chlorite, and montmorillonite), iron (hematite, magnetite, goethite, and nontronite), carbonate (dolomite and calcite), etc.

This paper focuses on characterization of structural features and composition of raw and acid-treated saponite clays originating from the Tashkiv deposit, in the Khmelnitsky region of Ukraine.

The unit cell of saponite consists of an octahedral sheet sandwiched between two opposing tetrahedral sheets with a negatively charged surface. Meanwhile, the interlayer space between these units is occupied by water molecules and cations, which may easily be exchanged with other metal cations to compensate for the negative charge of the unit cell's surface [18].

In saponite, the main cation of the tetrahedral sheet is $Si^{4+}$, but other cations such as $Al^{3+}$ and $Fe^{3+}$ have also been identified [18,19]. Substituting $Al^{3+}$ or $Fe^{3+}$ with $Si^{4+}$ within the tetrahedral sheet contributed to the increase of the layer's charge. Magnesium-rich saponite usually contained a $Mg^{2+}$ cation within its octahedral sheet. However, Fe-rich varieties with $Fe^{3+}$ and $Fe^{2+}$ substituted for $Mg^{2+}$ were also widespread in nature [18,20]. Substitutions of $Mg^{2+}$ with $Al^{3+}$ and $Ti^{4+}$ cations were also identified. Isomorphic forms of saponite differed in ion exchange and thermal properties [18,20]. Adsorption capacity values obtained from the analyses of raw saponite samples varied from 70 to 120 meq/100 g [13,21].

According to the results of an X-ray diffraction study [9], unit cell parameters of saponite from Orrock (Scotland) were a = 0.53 nm, b = 0.92 nm, and c = 1.49 nm. Results obtained by Suquet et al. [10] for saponite from Kozakov (Czech Republic) were slightly different—a = 0.53 nm, b = 0.92 nm, and c = 1.54 nm. The unit cell edge length in the c-direction for the latter differed due to cations exchanged within the interlayer space and its water content. The thickness of the layer may also vary from 1.2 to more than 1.5 nm [1].

Specific surface area of raw saponite collected from different deposits determined using nitrogen adsorption ranged from 30 to 70 $m^2g^{-1}$ [5,18,22]. A considerably higher BET specific surface area of 202 $m^2\ g^{-1}$ was determined for a raw sample of saponite from Vicalvaro (Spain) [13]. The authors attributed this to the abundance of small-sized particles of raw saponite, which originated from sediments.

## 2. Materials and Methods

Saponite clay samples were collected from the Tashkiv deposit (Khmelnitsky region, Ukraine). Acid-modified saponite was obtained by treating five grams of the raw powder clay under mechanical stirring with 150 mL of 2 M $H_2SO_4$ for four hours at 90 °C. Subsequently, the solid phase was washed and separated from the solution and dried at room temperature.

Structure and surface morphology of the raw and modified saponite were characterized using scanning electron microscopy with a focused ion beam (SEM/FIB Quanta 200 3D FEG), under the following analytical conditions—magnification = $200 - 350,000\times$, WD = 9.5–10 mm, HV = 20.0 kV, spot = 2.5, and signal = SE. Elemental composition (energy dispersive X-ray (EDX) elemental analysis) and elemental distribution (EDX elemental mapping) were characterized using a scanning electron microscope coupled with energy dispersive X-ray spectroscopy (SEM/EDX), under the following analytical conditions—magnification = $500\times$, WD = 25.0 mm, HV = 28.0 kV, spot = 3.0, and signal = SE. Transmission electron microscopy (TEM) (Tecnai F20 X-Twin) at an accelerating voltage of 200 kV, using HAADF (high angle annular dark field) and EDX (energy dispersive X-ray spectrometer) detectors, was applied to characterize the structure and surface morphology of saponite on the nanoscale.

X-ray diffraction (XRD) patterns were obtained using an X'Pert Pro diffractometer, in Cu K$\alpha$ radiation ($\lambda$ = 0.15406 nm; 60 kV, 55 mA) mode. Diffraction angle values were scanned from 5° to 120° (with a 2θ step size of 0.001). Mineral composition was determined by comparing the

experimental spectra to the reference powder diffraction patterns from the database integrated within the PANalytical's software package.

Porous structure parameters of the saponite samples were determined using the low-temperature (77 K) nitrogen adsorption/desorption method. Samples were degassed for 20 h at 453 K in vacuum. Adsorption data were obtained using Autosorb 6B, version 3.0 (Quantachrome Instruments, Boynton Beach, FL, USA). Specific surface area (S) was calculated from the Brunauer–Emmett–Teller (BET) adsorption isotherms and computational methods (DFT). The total pore volume ($V_{total}$) was determined by converting the volume of nitrogen adsorbed at $p/p_s$ = 0.99 to a volume of liquid adsorbate. The mesopore volume ($V_{mesBJH}$) and the pore diameter ($D_{BJH}$) were obtained using the Barrett–Joyner–Halenda (BJH) method [23]. The t-plot method [24] was applied to determine the micropore volume ($V_{mic}t$), the external surface area ($S_{ext}t$), and the micropore surface area ($S_{mic}t$). The Horvath–Kawazoe (HK) model [23] was used for micropore size calculations. Pore size distribution was determined using the standard non-local density functional theory (NLDFT) equilibrium model [25].

Fourier transform infrared spectroscopy (FTIR) spectra were recorded using a Spectrum 2000 (SMART Collector) spectrophotometer in order to determine the structural groups of the saponite samples. Each spectrum scan was recorded at a resolution of 8 $cm^{-1}$. FTIR spectra were recorded from 400 to 4000 $cm^{-1}$ using the KBr pellet technique in a ratio of sample to KBr of 1 to 40.

## 3. Results and Discussion

### 3.1. Elemental Composition of Saponite Samples

The elemental composition of the saponite samples obtained using SEM-EDX is summarized in Table 1. EDX spectra have shown that oxygen, iron, magnesium, aluminum, and silicon are the base elements of the raw samples with about 1% of titanium present. It should be noted that raw saponite is characterized by a considerable content (19.32%) of iron. Iron can be included in the saponite structure in both the octahedral and tetrahedral sheets or could also be present as a charge-balancing cation in a clay exchange complex [26,27].

**Table 1.** Elemental composition of saponite samples (values are expressed in mass%).

| Sample | C | O | Mg | Al | Si | K | Ca | Ti | Fe |
|---|---|---|---|---|---|---|---|---|---|
| Raw | 0.43 | 42.69 | 5.67 | 7.94 | 20.89 | 0.51 | 1.47 | 1.07 | 19.32 |
| Acid-treated | 1.48 | 48.36 | 2.39 | 9.48 | 30.19 | 0.59 | 0.68 | 0.92 | 5.63 |

After acid treatment, the content of Si and Al increased, while the content of Fe, Mg, and Ca decreased. According to the previously reported findings [28], treatment of clays using mineral acids of medium strength (2–4 M) led to leaching of exchangeable cations of interlayer space ($Na^+$ and $Ca^{2+}$) and cations of the octahedral lattice ($Al^{3+}$, $Fe^{3+}$, and $Mg^{2+}$) destroying packages in the process, considerably expanding the pore space and increasing the specific surface area. Meanwhile, the content of potassium was nearly unchanged. According to Stucki et al. [27,29] the fixation of $K^+$ in the $Fe^{3+}$-containing smectite octahedral sheet was possibly due to its reduction.

Acid treatment resulted in an easier removal of $Fe^{3+}$ cations than of $Mg^{2+}$ cations. This was corroborated by the decrease of Fe/Mg ratio in the modified samples. However, after acid treatment when the exchangeable cations of interlayer space have been removed, a considerable amount of $Fe^{3+}$ was still present in the solids. This suggests that $Fe^{3+}$ cations remain in the octahedral sheet of the clay structure [30].

Elemental distribution in the clay samples was assessed using SEM-EDX mapping techniques, and spot-mode quantitative analysis on points within single clay flakes (Figure 1). This mineral exhibits homogeneous distribution of titanium oxides in regular oval forms with different submicrometer sizes (mainly between 0.1 and 1 μm) which corresponds to anatase, according to data complemented by

XRD analysis (see Section 3.3 below). Non-isometric iron particles with sizes between 0.1 and 4 μm are uniquely distributed in the clay matrix.

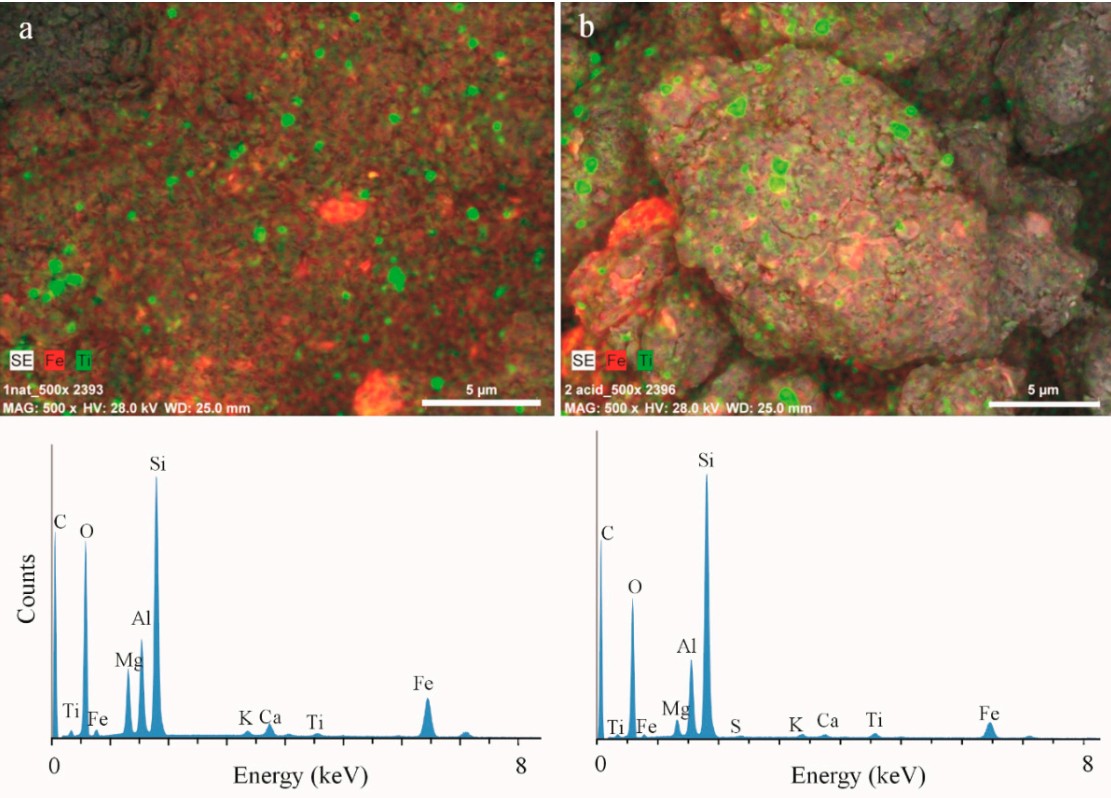

**Figure 1.** SEM photomicrographs and energy dispersive X-ray (EDX) spectra of saponite samples—(**a**) raw and (**b**) acid-treated.

### 3.2. Morphology and Structure of the Saponite Particles

The morphology and structure of the saponite particles before and after modification were determined using SEM (Figure 2a,b) and TEM (Figure 3a,b). Differences in the microstructure between raw and modified forms of the clay are visible from the SEM photomicrographs. For untreated saponite, the amorphous mass that consists of lath-like particles without well-defined borders are characteristic of the morphology of this clay mineral [13]. We cannot observe the individual crystallites, but only aggregates of irregular clay particles. Such cluster structures of clay do not allow determination of the size of individual crystallites. Saponite treated with acid exhibits smaller aggregates with dominant flake-like forms and irregular edges. The diameter of these aggregates is large (up to 10 μm), but the individual plate width is much less than 1 μm. These morphological changes are a result of the transformations in the clay matrix and at the boundaries between clay layers. Acid treatment results in modification of the surfaces of clays by disaggregation of particles, possible elimination of mineral impurities, and removal of exchangeable cations [31].

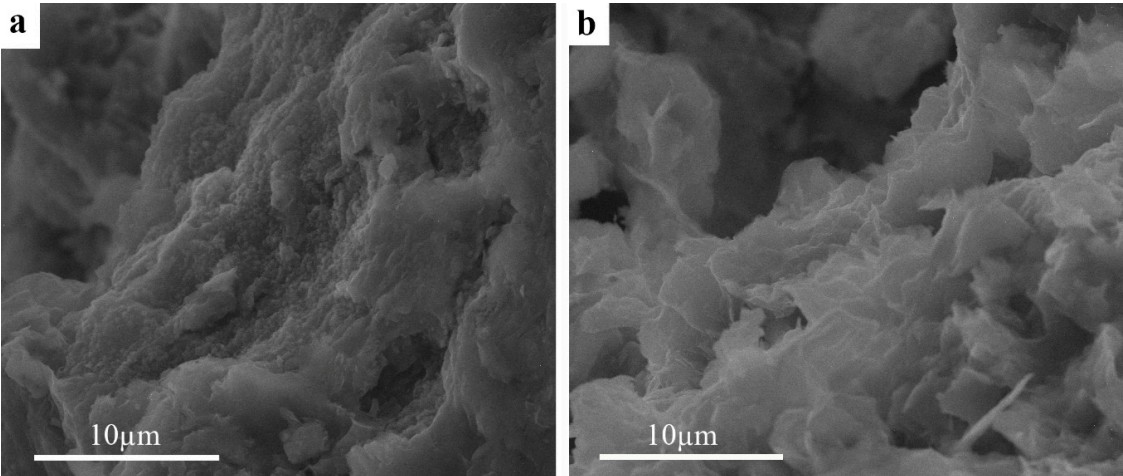

**Figure 2.** SEM photomicrographs of saponite—(**a**) raw and (**b**) acid-treated.

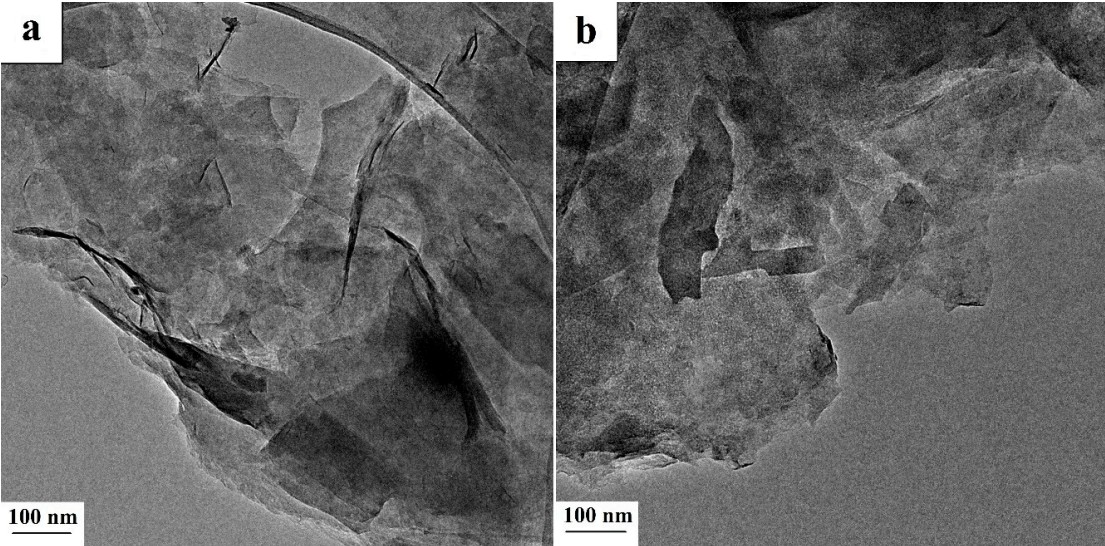

**Figure 3.** TEM images of crystalline features of saponite—(**a**) raw and (**b**) acid-treated.

Saponite imaged with TEM (Figure 3) was characterized by thin elongated plate-like particles of clay crystallites, single plates, and layered aggregates 10–200 nm long and 10–100 nm wide. The thicknesses of the particles did not exceed a few nanometers (2–5 nm). Aggregation of plate crystallites with flake-like forms of far smaller sizes had also been observed. TEM photomicrographs revealed the sites with irregular small clay aggregates (<10 nm) composed of even smaller individual clay nanocrystallites. It can also be observed that the saponite treated with acid was more dispersed in comparison to the raw sample. Mineral impurities such as quartz, hematite, and nontronite were revealed within samples that have been analyzed using TEM-EDX. Quartz was indicated in samples by columnar nanosized crystals 20–70 nm in the longest direction (Figure 4a). The impurities of hematite were present in two forms (Figure 4b,c). The first represented isometric particles with a size of about 10 nm. The second was present in circular arrangements of bladed crystals. In Figure 4d, agglomerates of fibrous nanocrystals of nontronite with a thickness of around 3 nm are depicted.

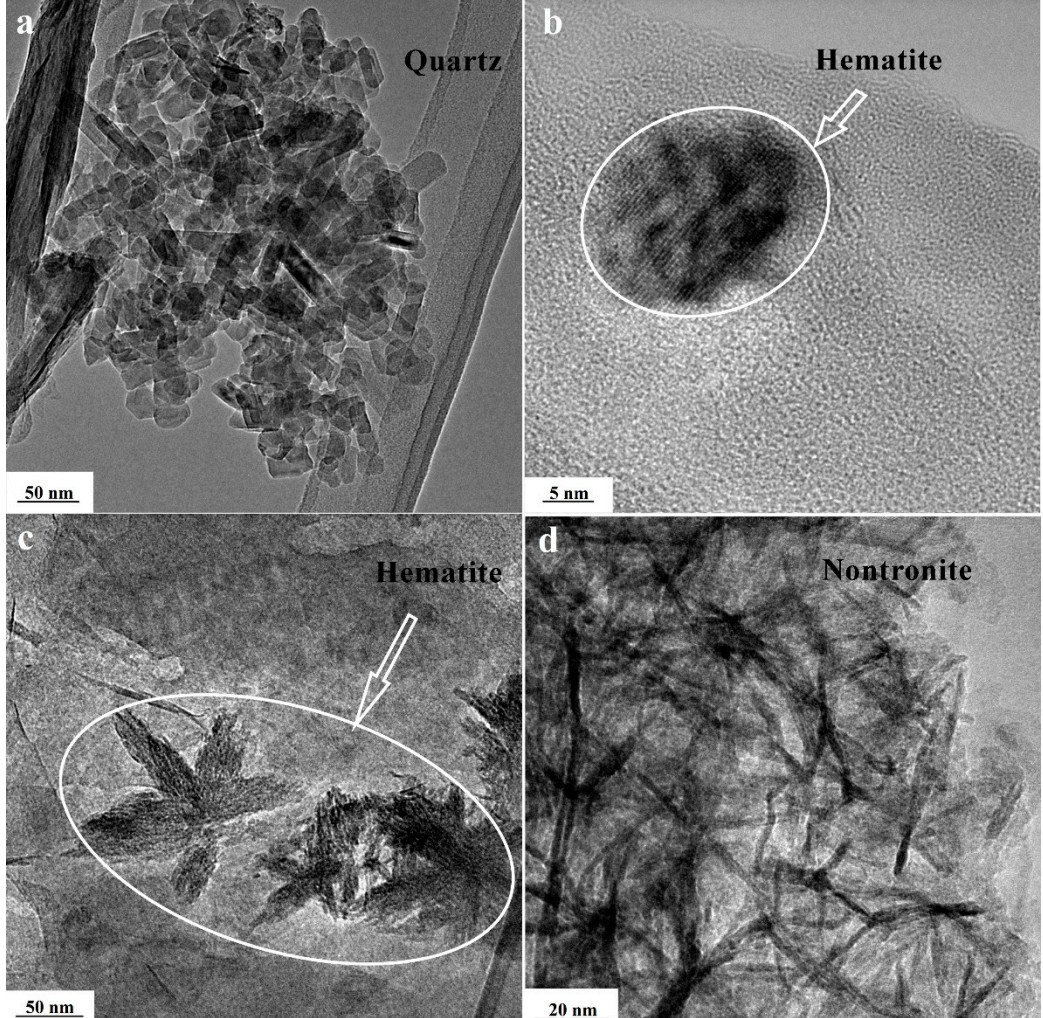

**Figure 4.** TEM images of mineral inclusions—(**a**) quartz; (**b**) hematite; (**c**) hematite; and (**d**) nontronite.

### 3.3. X-Ray Diffraction Patterns

X-ray diffraction patterns of the raw clay and acid-modified samples are shown in Figure 5. Diffractogram of the raw sample indicated that saponite was the dominant clay mineral. The most intense reflection appeared at $2\theta = 6.02°$ and corresponded to basal spacing of the c-axis ($d_{001}$) of the saponite cell with a value of 1.509 nm. For acid-treated saponite, a slight shift of this characteristic reflection was observed, related to the decrease of basal spacing to 1.463 nm. The decrease of interlayer space for the acid-treated saponite was due to cation exchange (Fe, Ca, and Mg) with smaller hydronium cations during acid treatment. The broad form of these reflections suggested low crystallinity and small particle size of the samples [13]. A small, but rather sharp reflection near $2\theta = 26.5°$ ($d_L = 0.334$ nm) corresponded to a substantial content of quartz in the samples. Other small reflections detected in the X-ray diffraction patterns corresponded to the relative concentration of crystalline impurities contaminating the raw sample, mainly anatase ($d_L = 0.368$ nm) and hematite ($d_L = 0.352$ nm). After acid treatment, the intensity of the reflections assigned to mineral admixtures clearly increased (especially for quartz).

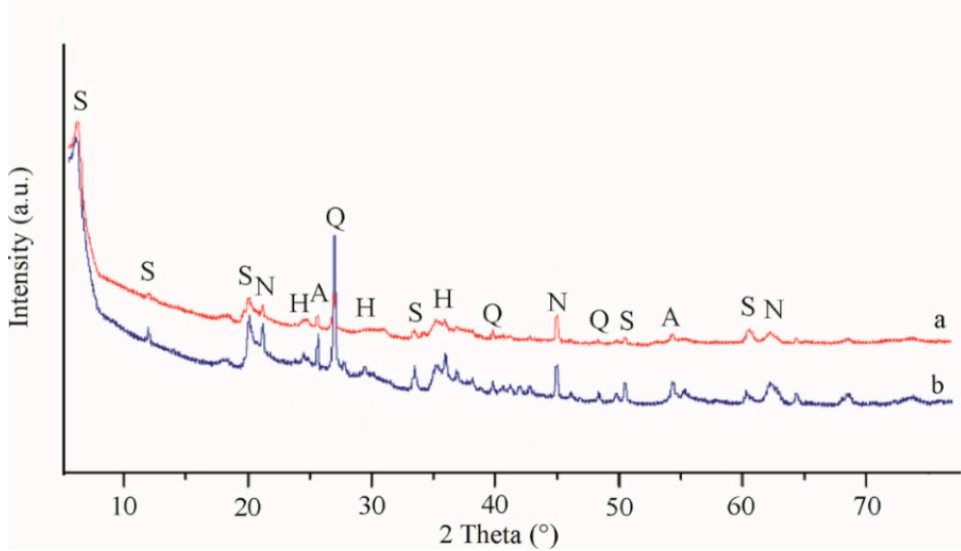

**Figure 5.** X-ray diffraction patterns of the studied saponite samples—(**a**) raw saponite and (**b**) acid-treated saponite. S, saponite; Q, quartz; N, nontronite; H, hematite; and A, anatase.

The reflection at $2\theta = 60.04°$ ($d_L = 0.1535$ nm) corresponded to trioctahedral structure of the saponite phase. Presence of the dioctahedral clay phase as nontronite could be related to the reflection at $2\theta = 62.51°$ ($d_L = 0.1497$ nm) [20,32].

### 3.4. Nitrogen Adsorption/Desorption Isotherms

Adsorption–desorption isotherms of nitrogen for saponite samples (Figure 6) were similar to type IV of the Brunauer–Deming–Deming–Teller (BDDT) classification with the hysteresis loop of the type H3 of IUPAC classification, that had often been associated with slit-shaped pore structures [33]. On the other hand, the H3 loop did not have a plateau at high p/p0 values and the whole adsorption branch of the hysteresis loop followed the same path as the corresponding part in a Type II isotherm. According to the opinion of Sing and Williams [34] such isotherms of II pseudo-type character were due to the metastability of the adsorbed multilayer. This was associated with the low degree of pore curvature and non-rigid, slit-shaped pores structure, or non-rigid assemblages of platy particles. Rouquerol et al. [35] also considered that this type of isotherm can be referred to as a Type II (b) with the H3 hysteresis loop, but this hypothesis has not yet been generally accepted.

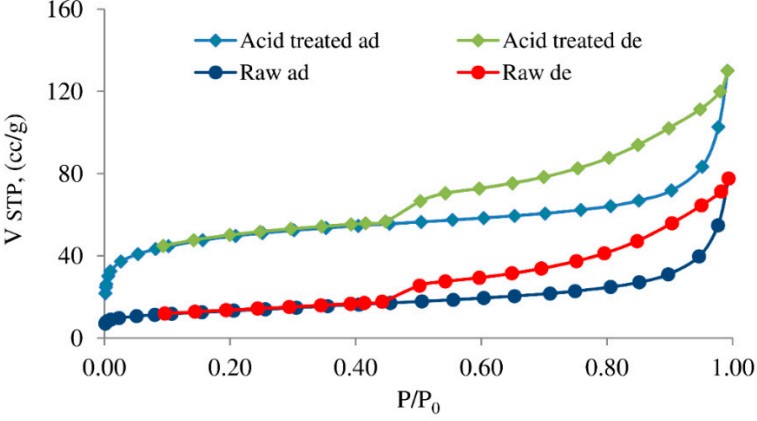

**Figure 6.** Nitrogen adsorption/desorption isotherms for saponite samples.

The shapes of the saponite samples isotherms indicated the presence of both micro and mesopores. The hysteresis loop began at a relative pressure near 0.4 and corresponded to the loading and unloading of mesopores [36]. The isotherm curves sharply raised at the beginning at a small value of relative pressure ($p/p_0 < 0.1$) that was caused by the presence of free micropores in the adsorbent samples in which nitrogen adsorption occurred. Values of several important parameters of the pore structure obtained from the analyses of the isotherms are summarized in Table 2.

**Table 2.** The porous structure parameters of saponite samples.

| Sample | Specific Surface S, $m^2\ g^{-1}$ | | | | Pore Volume V, $cm^3\ g^{-1}$ | | | | Average Pore Diameter D, nm |
|---|---|---|---|---|---|---|---|---|---|
| | $S_{DFT}$ | $S_{ext}t$ | $S_{mic}t$ | $S_{BET}$ | $V_{total}$ | $V_{mesBJH}$ | $V_{mic}t$ | $V_{micHK}$ | $D_{DFT}$ |
| **Raw** | 47 | 31 | 16 | 47 | 0.134 | 0.119 | 0.008 | 0.028 | 5.3 |
| **Acid-treated** | 189 | 38 | 139 | 177 | 0.201 | 0.130 | 0.062 | 0.086 | 1.4 |

Specific surface area of the saponite sample after acid treatment significantly increased from 47 to 189 $m^2\ g^{-1}$, while the total pore volume increased from 0.134 to 0.201 $cm^3\ g^{-1}$. The considerable increase in the specific surface area was due to the increase in internal surface area which was equivalent to the micropore surface area. The micropore volume for the saponite treated with acid increased seven-fold, up to 0.062 $cm^3\ g^{-1}$ and 0.086 $cm^3\ g^{-1}$, as calculated from the t-plot and the Horvath–Kawazoe model, respectively. These results are in accordance with analyses of saponite samples originating from deposits from other locations [18,37].

Increase in pore volume was due to the removal of interlayer cations ($Fe^{3+}$, $Mg^{2+}$, and $Ca^{2+}$) that caused weakening of bond between the layers and the expansion of space between the clay particles [31].

Analysis of the pore size distribution using the NLDFT model (Figure 7) provided pore diameter maxima with a wide peak that corresponded to 5–15 nm pores. Pore size distribution of the saponite treated with acid was different from the one obtained for the raw sample. A few narrow peaks that correspond to a considerable increase of pores with sizes ranging from 1 to 3 nm and from 5 to 6 nm were observed.

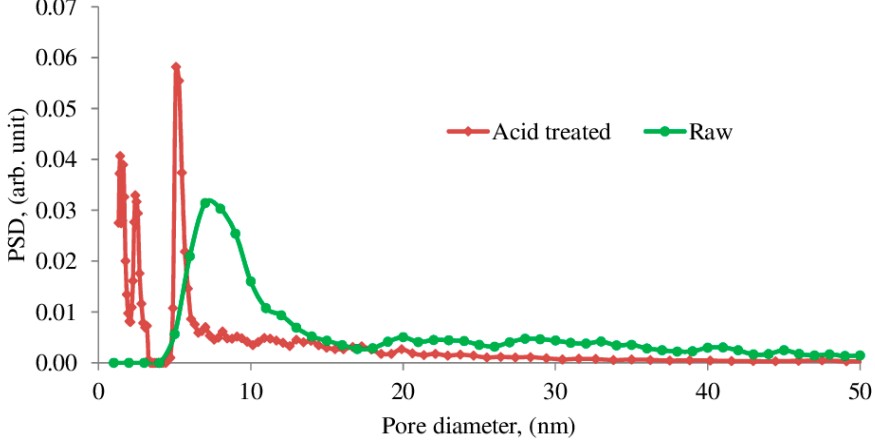

**Figure 7.** Pore-size distribution by application of the non-local density functional theory (NLDFT) model.

*3.5. Fourier Transform Infrared Spectra of Saponite Samples*

Fourier transform infrared spectra (Figure 8) indicated the existence of different types of structural bonds and functional groups for saponite samples. The high frequency region of the spectrum contained an overlapping wide absorption band in the 3650–3000 $cm^{-1}$ range, which corresponded

to stretching vibration of hydroxyl groups [19,38]. FTIR spectra of raw saponite showed an intensive band at about 3623 cm$^{-1}$, which corresponded to stretching vibration of OH groups within Mg–OH–Fe bonds in trioctahedral sheets [39]. The absorption band at about 3574 cm$^{-1}$ could be attributed to OH stretching vibration of Fe–OH–Fe bonds and indicated the isomorphic substitution of Fe by Mg in the trioctahedral sheets of Fe saponite [38,40,41]. According to the findings of Mas et al. [39] this spectral region (3650–3500 cm$^{-1}$) corresponding to OH stretching was related to the co-existence of Fe- and Mg-rich phases in octahedral sheets of clay minerals. Absorption in a wide spectral range of 3500–3000 cm$^{-1}$ with two maxima at 3400 cm$^{-1}$ and 3270 cm$^{-1}$ could be assigned to OH stretching vibrations of hydrogen-bonded water molecules in the interlayer region [38,41,42].

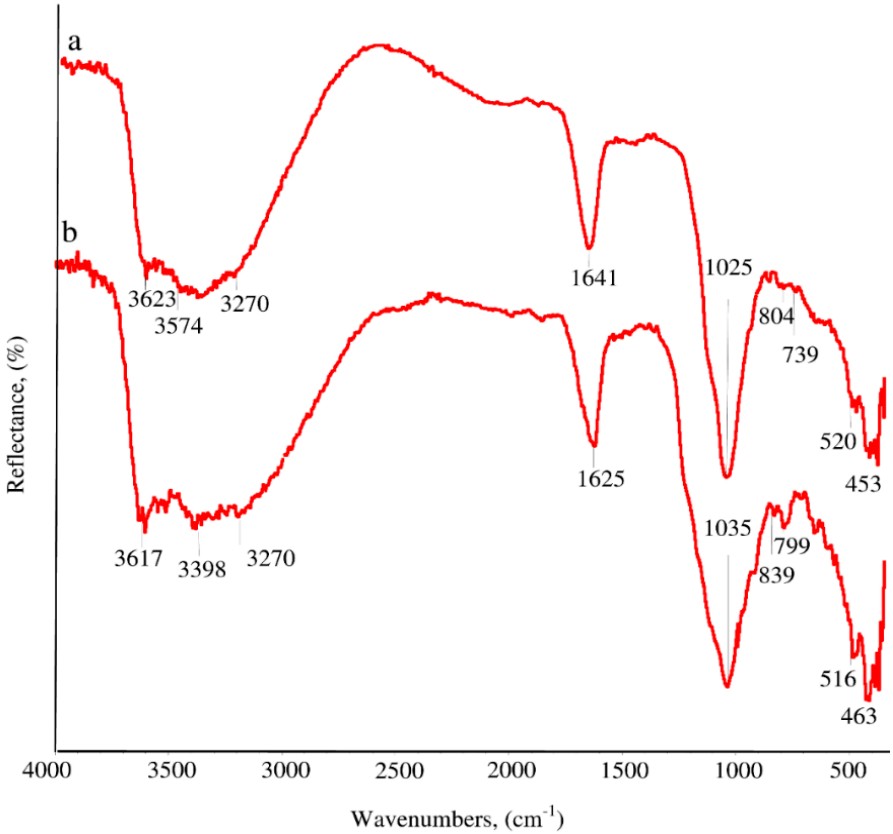

**Figure 8.** Fourier transform infrared spectra of raw (**a**) and acid-treated saponite (**b**).

The absorption band at 1641cm$^{-1}$ was related to the deformation of molecular water H–O–H [20]. An intense absorption band with a maximum around 1025 cm$^{-1}$ corresponded to stretching of O–Si–O comprising the tetrahedral sheet [42]. Absorption bands at 839, 657, and 453 cm$^{-1}$ were due to Fe–OH–Fe, Fe–O out of plane, and Fe–O–Si bending vibration, respectively. This confirmed the high Fe-content within the octahedral sheet [43]. A small peak at 520 cm$^{-1}$ was attributed to the Si–O–Al deformation and indicates the presence of some aluminum in the tetrahedral sheet [20].

FTIR spectra of samples treated with acid depict changed in saponite structure. The O–Si–O band changed in form and had exhibited a shift from 1025 to 1035 cm$^{-1}$, while the OH bending band intensities decreased. This pointed to a partial alteration of the clay structure [22]. The intensities of the bands at 658 cm$^{-1}$ (Fe–O bonds), 3574 cm$^{-1}$ (Mg–OH–Fe bonds), and 3623 cm$^{-1}$ (Al–OH–Mg bonds) decreased after acid treatment upon removal of Mg$^{2+}$ and Fe$^{3+}$ cations in agreement with SEM-EDX data. The absorption band at 1641cm$^{-1}$ was attributed to the deformation of molecular water, which was maintained after the treatment exhibiting a small decrease and a shift to 1625 cm$^{-1}$. However, the acid treatment caused the most intense bands at 799 cm$^{-1}$, 516 cm$^{-1}$, and 463 cm$^{-1}$ to appear. These bands were attributed to Si–O bonds of the free amorphous silica phase formed as a result of

the partial destruction of tetrahedral sheets [44], Si–O–Al and Si–O–Si bonds due to dissolution of the octahedral sheet [45] by leaching of Mg and Fe, respectively.

## 4. Conclusions

Results of this study have shown that raw saponite forms amorphous aggregates of flake-like or lath-like clay crystallites without well-defined borders. Disaggregation of clay particles, as a result of acid treatment, prompted the formation of even smaller clay crystallites with dominant irregular flake-like forms, which corresponded to the elimination of exchangeable cations from the interlayer space and the weakening of bonds between the clay crystallites.

Raw saponite is composed predominantly of a trioctahedral smectite, while an admixture of dioctahedral phase corresponds to a substantial quantity of nontronite and a certain amount of associated minerals such as quartz, anatase, and hematite. Submicrometer anatase particles (1.1% of titanium) are homogeneously dispersed in the clay matrix with >1%. The high content of iron (19.3%) is distributed in the saponite clay structure within octahedral and tetrahedral sheets, and exchangeable cations in the interlayer space, or as a non-isometric form of micrometer-sized hematite particles.

Results of nitrogen adsorption/desorption revealed that the porous structure of saponite is comprised of micropores and mesopores in the form of narrow slits. Treatment with acid resulted in a considerable increase of pores with size ranging from 1 to 6 nm and an increase in the specific surface area of saponite ranging from 47 $m^2$ $g^{-1}$ to 189 $m^2$ $g^{-1}$, while the total volume within the pores increased from 0.134 $cm^2$ $g^{-1}$ to 0.201 $cm^2$ $g^{-1}$.

Spectroscopic studies revealed the existence of different types of hydroxyl groups associated with dioctahedral and trioctahedral sheets of raw and modified saponite. Treatment with acid led to the partial decrease in the number of hydroxyl groups which were derived from octahedral sheets.

Results obtained in this study are of interest from practical point of view, since the textural characteristics and chemical composition of saponite founded in Khmelnitsky region might be useful for formulation of industrial materials. They may cause noticeable effects on quality of industrial products such as catalysts, adsorbents, sealants, adhesives, etc.

**Author Contributions:** Conceptualization, A.G.; formal analysis, H.S.; funding acquisition, H.S. and M.S.; investigation, H.S. and A.G.; methodology, M.S. and A.G.; project administration, B.B.; supervision, M.S. and B.B.; writing—original draft, H.S.; Writing—review and editing, M.S. and V.R.

**Funding:** This research was funded by the International Visegrad Fund, grant number 51501546/2015.

**Conflicts of Interest:** The authors declare no conflict of interest. The funders had no role in the design of the study; in the collection, analyses, or interpretation of data; in the writing of the manuscript, or in the decision to publish the results.

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
