# Peer review of "Structural, Mineral and Elemental Composition Features of Iron-Rich Saponite Clay from Tashkiv Deposit (Ukraine)"

_colloids, doi:10.3390/colloids3010010_

Round 1
Reviewer 1 Report
This paper presents the thorough characterization of structural features and composition of raw and acid treated saponite clays originating from the Tashkiv deposit (Khmelnitsky region, Ukraine).These issues are important and interesting however some remarks concerning the preparation of the manuscript should be taken into consideration while revising the paper:
Can the authors explain the likely cause of a significant reduction in pore size while increasing Vp for the acid treated sample (Table 2)?
Table 2 and text (lines 244-253): Units should be listed in accordance with MDPI requirements.
line 199: The shapes of the saponite sample isotherms indicate….. sample should be used in plural because the authors describe 2 samples.
line 262: What is the reason for repeating the content of TiO2 two times: Submicrometer anatase particles (1.1 % of titanium) are homogeneously dispersed in the clay matrix with > 1 % of titanium.
In my opinion, some sentences should be corrected:
line 251: These bands are attributed to Si–O bonds of the free amorphous silica phase formed as a result of the 251 partial destruction of tetrahedral sheets [44] Si–O–Al and Si–O–Si bonds due to dissolution of the 252 octahedral sheet by leaching of Mg and Fe, respectively [45] and resulting.
line 255: Results of this study have shown that raw saponite is forms amorphous aggregates of flake-like or lath-like clay crystallites without well-defined borders.
Author Response
Point 1. Can the authors explain the likely cause of a significant reduction in pore size while increasing Vp for the acid treated sample (Table 2)?
Answer: Significant reduction of average pore diameter(D) is caused by a vast increase the contribution of micropores (d< 2 nm) for the saponite treated with acid. This is clearly visible in the pore size distribution obtained by the NLDFT method (figure 7).
Point 2. Table 2 and text (lines 244-253): Units should be listed in accordance with MDPI requirements.
Answer: The appropriate corrections have been done.
Point 3. line 199: The shapes of the saponite sample isotherms indicate….. sample should be used in plural because the authors describe 2 samples.
Answer: The appropriate corrections have been done.
Point 4. line 262: What is the reason for repeating the content of TiO2 two times: Submicrometer anatase particles (1.1 % of titanium) are homogeneously dispersed in the clay matrix with > 1 % of titanium.
Answer: The appropriate corrections have been done.
Point 5. In my opinion, some sentences should be corrected:
line 251: These bands are attributed to Si–O bonds of the free amorphous silica phase formed as a result of the 251 partial destruction of tetrahedral sheets [44] Si–O–Al and Si–O–Si bonds due to dissolution of the 252 octahedral sheet by leaching of Mg and Fe, respectively [45] and resulting.
line 255: Results of this study have shown that raw saponite is forms amorphous aggregates of flake-like or lath-like clay crystallites without well-defined borders.
Answer: The appropriate corrections have been done.
Reviewer 2 Report
The authors charaterized the sturcture, composition and morphology of saponite clay in Ukraine, which is a rarely explored material, using various methods including BET, SEM, TEM, XRD and IR. They compared with consequence of acid treatment on these surface/interface properties. The experiment is kind of regular and easy. But I feel this is a well-written manuscript. The results makes sense to readers and the science presented here are of interest to readers of colloids and interfaces.
I would recommed for publication with present form
Author Response
Thank you for your revision.
Reviewer 3 Report
Please see the attached file.

Author Response
Thank you for your revision and valuable comments.
Point 1. Introduction, experiments, results and discussion, and conclusions are presented. The number of references is sufficient.
Answer: Thank you for your comments.
Point 2. In the section Materials and Methods, please follow sequence (as used in results) for description of SEM, EDX, TEM, XRD, Nitrogen adsorption/desorption isotherms, and FTIR.
Answer: The necessary changes have been done.
Point 3. For text in lines 94-103, the line-spacing seems different than the remaining the text.
Answer: It has been changed.
Point 4. For electron microscopy analysis please include some (brief) additional instrument details such as operating voltages of SEM and TEM, Cs and optimum defocus of the TEM, type of imaging mode: bright filed or dark field.
Answer: These additional instrument details have been added to the section "2. Materials and Methods".
Point 5. Line 121: Authors state that ...by the decrease of Mg/Fe ratio...... Actually from the Table 1, before acid treatment the Mg/Fe ratio is 0.29 while after acid treatment it is 0.42. So it should be .....increase of Mg/Fe.....?
Answer: You are right, in manuscript was mistake, “Mg/Fe” has been changed for “Fe/Mg”. After acid treatment “Fe/Mg” ratio decrease from 3.40 to 2.35, due to easier removal of Fe3+ cations than of Mg2+.
Point 6. Line 153-157 Authors state that "The thicknesses of the particles do not exceed a few nanometers (2-5 nm). TEM photomicrographs reveal the sites with irregular small clay aggregates (< 10 nm) composed of even smaller individual clay nanocrystallites." 2 Is there any TEM experimental proof for this statement? Please indicate diagram or part of diagram as evidence.
Answer: Unfortunately, we do not use the special software (e.g. ImageJ) to perform measurement of particle size. It is a good option for clearly separated particles with a regular shape, but in our case samples have complex morphology with an irregular shape, for this reasons accurate size statistics are more complicated. Measurement of particle size was performed using the ruler in manual mode.
Point 7. Line 162 Typographical error? Should be 10 nm instead of 10 μm?
Answer: This mistake has been corrected, “μm” has been changed for “nm”.
Point 8. Line 167 state that "X-ray diffraction patterns of the raw clay, and acid modified samples are shown in Figure 5." Since intensity of XRD data of raw and acid-treated are directly compared was background correction of XRD data performed?
Answer: The intensity of XRD patterns of the raw and acid-treated saponite samples were compared with using diffractograms’ background correction.
Point 9. If possible please enlarge figures 5. It is would be earlier to clearly see shifts and change in intensities.
Answer: It has been enlarged.
Point 10. Line 178-179 Authors say that "After acid treatment, the intensity of the reflections assigned to mineral admixtures 178 clearly increased (especially for quartz)". Please explain briefly the reasons? Are the mineral admixtures more chemically stable?
Answer: The mineral admixtures such as quartz and anatase more chemically stable to acid treatment than saponite. The reason for the lower reactivity of mineral admixtures its macromolecular structure.
Point 11. If possible please enlarge figures 8 (especially horizontally).
Answer: It has been enlarged.
Point 12. Line 96 state that each FTIR spectrum scans was recorded at a resolution of 8 cm−1 . Is it possible for authors to use higher resolution? Authors have reported shifts in FTIR peak positions of ~10 cm-1 for acid treated saponite clays (example line for O-Si-O bonds as reported on line 243-244). Higher resolution data acquisition could reveal even smaller peak shifts and smaller changes in the FTIR spectrum in more details?
Answer: You are right, the higher resolution data acquisition would be more accurate. But in our opinion the FTIR spectrum scans would be have similar trend in general.